# The Mediating Effects of Marital Intimacy and Work Satisfaction in the Relationship between Husbands’ Domestic Labor and Depressive Mood of Married Working Women

**DOI:** 10.3390/ijerph17124547

**Published:** 2020-06-24

**Authors:** Su-Yeon Choi, Hyoung-Ryoul Kim, Jun-Pyo Myong

**Affiliations:** 1Department of Seoul Atopy Asthma Education Information Center, Seoul Medical Center, Seoul 02053, Korea; 2512happy@naver.com; 2Department of Occupational and Environmental Medicine, Seoul St. Mar’s Hospital, College of Medicine, The Catholic University of Korea, Seoul 06591, Korea; cyclor@catholic.ac.kr

**Keywords:** husbands’ domestic labor, marital intimacy, work satisfaction, depressive mood, mediating effect

## Abstract

The purpose of the study was to examine the relationships between the husbands’ domestic labor and marital intimacy, work satisfaction, and depressive mood in married working women. We used the sixth (2016) dataset from the Women and Families Panel Survey conducted by the Korean Women’s Development Institute (KWDI). The subjects were 791 married working women who lived with a wage-earner husband and who did not have a housework assistant. The correlations between variables were measured and the fit of the structural equation model was assessed. We used a mediation model in which the husbands’ domestic labor affected the depressive mood of married working women through mediation of marital intimacy and work satisfaction. Bootstrapping was used to verify the significance of the indirect effects of the mediating variables. Husbands’ domestic labor had a significant effect on married women’s marital intimacy and work satisfaction, but no significant direct effect on depressive mood. Marital intimacy had a significant effect on work satisfaction, and these two variables were significantly related to reductions in the depressive mood score. Husbands’ domestic labor was found to be a complete mediator of depressive mood through its effects on marital intimacy and work satisfaction. Husbands’ domestic labor did not directly reduce married working women’s depressive mood scores, but instead reduced them indirectly through effects on marital intimacy and work satisfaction.

## 1. Introduction

In Korea, a growing number of families are diverging from the traditional family structure where men are the breadwinners to a new gender role structure where housework and parenting is divided and both men and women are earners. However, according to Statistics Korea data from 2015 and Ministry of Employment and Labor data from 2017, the division of roles in reality does not seem simple because housework and parenting are generally still considered the woman’s responsibility [1,2]. Even when compared to neighboring countries, it seems difficult to balance work and family among dual-income Korean women [3,4]. For dual-income couples, unless the traditional gender and family norms of housework and parenting change or a conflict occurs between the two areas, problems related to this may become inevitable [5,6]. The multiple roles of dual-income women can present a potential risk factor for physical and mental health. Research related is needed because this can directly or indirectly affect not only the quality of life, interpersonal relationships, and physical and mental health of the individual but also the community they belong to.

In married life, the spouse is the most important source of social support and the first person to seek when facing difficulties [7]. The practical support of husbands who share housework with wives (who bear multiple roles in work and family) can lighten women’s burdens and provide comfort and positive support for women’s physical and mental health. Marital intimacy, which is a byproduct of marital interaction, has been shown to reduce family and job stress by promoting the sharing of housework and parenting in dual-earner couples [8]. It can also reduce stress relating to the role of work and family [9]. Work satisfaction for female wage earners reflects the level of overall life satisfaction beyond the workplace and can be regarded as an important factor affecting a person’s psychological condition and territory [10,11]. In this study we focused on how husbands’ domestic labor, marital intimacy, and work satisfaction in women from dual-earner households were associated with depressive mood. This study aims to contribute to providing the basic data necessary to prepare women’s health policies and directions that can enhance the psychological well-being and mental health of women from dual-earner households.

## 2. Materials and Methods 

### 2.1. Study Population

The data of study used in the present report were from the sixth (2016) wave of the panel study initially started in 2007 by the Korea Women’s Development Institute (KWDI).

That wave consisted of 9997 women aged 19 to 64 years old. The subjects of study were female wage earners, including regular, temporary, and daily employees, who account for 77.2% of all female workers in Korea [12]. Of the total sample, participants were excluded from the current analyses if they were single, divorced, separated, or widowed (*n* = 8700); had a housework assistant (*n* = 89); were married to a husband who was not a wage earner (*n* = 412); and/or who had non-responses and missing values (*n* = 5) (Appendix A). 

### 2.2. Ethical Considerations

This study was approved by the Institutional Review Board of The Catholic University of Korea (IRB approval number: MC19ZESI0041).

### 2.3. Key Variables

#### 2.3.1. Husbands’ Domestic Labor

The measure of husbands’ domestic labor used in this study was “the degree of husband’s housework over the past month” as related to food preparation, dishwashing, laundry, grocery shopping, and house cleaning. This was assessed via husband and wife’s housework questions from the Women and Families Panel Survey. Each item was rated on a 6-point Likert scale ranging from “Almost every time (1)” to “Not at all (6)”. After accounting for reverse coding, the responses to each item were summed. Thus, a higher score signifies a greater contribution to housework.

#### 2.3.2. Marital Intimacy

Marital intimacy refers to the degree of shared cognitive, emotional, and sexual intimacy between a couple [13,14], and was measured by the responses to four questions from the Women’s Family Panel Survey about the frequency of conversation, similarity of views, marital sexual satisfaction, and trust. Each item was rated on a 4-point Likert scale ranging from “Always (1)” to “Never (4)”. After accounting for reverse coding, the responses to each item were summed. Thus, a higher score signifies higher marital intimacy.

#### 2.3.3. Work Satisfaction

Work satisfaction is a multi-dimensional concept that encompasses the overall work situation for wage earners [15]. Work satisfaction was assessed from the responses to ten questions from the Women’s Family Panel Survey about wage or income level, job security, work environment, working hours, the potential for individual development, workplace communication and personal relationships, benefits, recognition of performance, and overall job satisfaction. Each item was rated on a 5-point Likert scale ranging from “Very Satisfied (1)” to “Very Dissatisfied (5)”. After accounting for reverse coding, the responses to each item were summed. Thus, a higher score signifies greater work satisfaction.

#### 2.3.4. Depressive Mood

The depressive mood scale used in this study was a shortened form of the Center for Epidemiological Studies of Depression Scales (CES-D) 20, developed by the Center for Epidemiological Studies at the National Institute of Mental Health (NIMH) in the United States. The scale measures self-reported symptoms of depression over the past week [16,17], with a total of 10 questions. Each item was rated on a 4-point Likert scale ranging from 0 to 3 points and reverse coded except for the positive questions (questions 5 and 8). A higher score on the scale signifies a greater level of depressive mood.

#### 2.3.5. Control Variables

The control variables in this study included the demographic characteristics of the subjects. Factors that may affect the depressive mood of married working women include age, education, number of children, occupation, average monthly income, and husband’s average income [18,19,20]. Since economic factors can have a significant impact on depressive mood, we tested for this effect with logistic regression analysis.

### 2.4. Statistical Analysis

PASW 18.0 and IBM SPSS Amos 25.0 were used for the empirical analysis. Frequency analyses, descriptive statistics, reliability analyses, and correlation analyses were performed on the demographic characteristics of the subjects and to check the normality, reliability, and multicollinearity of the measurement variables. After identifying a correlation, a confirmatory factor analysis was performed according to the two-step approach proposed by Anderson and Gerbbing to estimate the structural equation model [21].

First, to verify the validity of the measurement model, confirmatory factor analysis of potential variables was conducted to confirm the convergent validity, discriminant validity, and the fitness index [22,23].

Next, the fitness of the structural equation model was verified using the maximum likelihood method. χ^2^, SRMR, RMSEA, and GFI were used for the absolute fit index; NFI, TLI, and CFI were used for the incremental fit index; and PRATIO, PNFI, and PCFI were used for the parsimonious fit index [23]. Once the conformity of the final determined model was confirmed, a structural model path analysis was performed to verify the paths between the major variables.

Finally, bootstrapping was performed to verify the mediating effects of the final model and the phantom variables were used to verify the statistical significance of the individual mediated effects. Since the accuracy of bootstrapping increases with the number of estimates, the number of estimates was set to 10,000.

## 3. Results

### 3.1. Descriptive Statistics

Demographic characteristics are shown in Table 1. The major variables had skewness values ranging from −0.637 to 2.711 and kurtosis values ranging from −0.606 to 8.353, but thus did not, for every variable, satisfy the assumptions for normality. The reliability of the measured variables was found to be satisfactory (Table 2). For the depressive mood scale, Cronbach’s α was high after removing items 5 and 8. When analyzing the correlations between the main variables, we found statistically significant correlations between the latent variables except between husbands’ domestic labor and depressive mood. The correlations ranged from −0.140 to 0.278, so there was no multicollinearity (Table 3).

### 3.2. Measurement Model Analysis

To test whether the measured variables were appropriately measuring the latent variables, we performed a confirmatory factor analysis and found that RMSEA and GFI did not reach the recommended level. To improve the fitness, we performed a reanalysis of the model in which the correlations between the errors of the measured variables that had modification indices of 20 or higher were determined one by one. When we evaluated the fit of the modified model, all the indices were found to be appropriate (RMR (0.019), RMSEA (0.037), GFI (0.943), NFI (0.951), CFI (0.974), TLI (0.970), PRATIO (0.875), PNFI (0.832), and PCFI (0.852)), except for the fitness reliability χ^2^ (Appendix A). 

Thus, the modified model was judged appropriate for the study. Factor loadings were above 0.50 for all sub-factors and were statistically significant. The average variance extracted (AVE) ranged from 0.517 to 0.800 and thus met the standard (>0.50), and the construct reliability (CR) ranged from 0.842 to 0.969 and thus met the standard (>0.70), securing convergent validity (Table 4) [24]. Furthermore, the AVE value was larger than the squared correlation coefficient between the latent variables obtained through confirmatory factor analysis, securing discriminant validity (Appendix A).

### 3.3. Structural Model Analysis

We performed a path analysis to verify paths between the major variables in the structural We performed a path analysis to verify paths between the major variables in the structural model and found that all path coefficients were significant except for “Husbands’ domestic labor → Depressive mood” (Table 5). To identify the optimal model, we analyzed the modified model, excluding the path that was not statistically significant. However, the fit index did not improve significantly, and the results of the analysis did not change much. Thus, the hypothetical model was adopted as the final model (Figure 1 and Appendix A). Bootstrapping was used to verify the mediating effects of the final model identified by the analysis of the structural model and to perform a decomposition of the total effect into direct effects and indirect effects.

### 3.4. Mediating Effects and Hypothesis Testing

The SPSS Amos program contains the limitation that only the overall indirect effect is provided when using the bootstrapping method with more than two mediation effects; thus, the significance and detailed mediating effects for each mediation variable cannot be verified [25,26]. To solve this problem, we set up a phantom variable. A phantom variable is a hypothetical variable that is set so as not to affect the model fitness or parameter values when verifying multiple mediated effects [27], and the significance of the indirect effects can be verified by analyzing the phantom variable with the bootstrapping method. The analysis showed that marital intimacy had a statistically significant effect on the relationship between husbands’ domestic labor and work satisfaction. Furthermore, the mediating effects of marital intimacy and work satisfaction on the relationship between husbands’ domestic labor and married working women’s depressive mood were statistically significant. 

In other words, since the 95% confidence interval value of the h3, h5, h7, and h9 paths did not include 0, the mediating effect of marital intimacy and work satisfaction was statistically significant at the *p* < 0.05 level. The results of the test for mediating effects are shown in Table 5.

## 4. Discussion

The purpose of the study was to examine the relationships between the husbands’ domestic labor and marital intimacy, work satisfaction, and depressive mood in married working women. 

First, we found that married women in their 40s to 50s often engaged in economic activities, mostly in professional and clerical work and in the form of irregular employment rather than regular employment. The dual-earner household rate is high for women in their 40s to 50s when the cost of children’s education is high, as explained by results of Statistics Korea where women were found to be involved in mostly temporary or part-time work [28,29]. In Korea, marriage and childbirth create large changes in the lives of women but less so for men. Although various social systems are designed for work–family balance, it can be difficult for women to return to their jobs after childbirth or to obtain a job with stable working conditions. Specific and practical social support and changes should be supported, as dual-income households are becoming a universal circumstance. 

Second, our results showed that husbands’ domestic labor had a statistically significant effect on married working women’s marital intimacy and job satisfaction. For dual-earner couples, spouse support is a highly important factor leading to a healthy family life [30,31], and this is consistent with previous studies showing that marital satisfaction increases as husbands’ domestic labor increases [32,33,34]. It is considered that double-earner husbands’ domestic labor efforts to balance work and family can serve as a factor to enhance marital intimacy since this effort is consoling to working women. In a previous study of women in dual-earner households, spouse support also revealed a static relationship with work satisfaction [35]. On the other hand, the effect on depressive mood was not statistically significant. 

According to previous studies, low income and economic stress increase depression [36,37]. This study confirmed whether the income earned by either the wife or the husband affected the depressive mood score, but no statistically significant results were found. Unlike previous studies that generally did not rule out negative effects of economic factors on depression, our study suggested that husbands’ domestic labor, rather than their wages, acts as a real or emotional support system for women who play multiple work and family roles. The level of depression of married working women is related to the relationship with the husband and satisfaction with their roles in the family or workplace [38].

Third, in terms of the relationship between the husbands’ contribution to housework and depressive mood, the mediating effect of marital intimacy and work satisfaction was found to be complete. In other words, although the husbands’ domestic labor did not directly affect the depressive mood of married working women, there were indirect effects via marital intimacy or work satisfaction, or both. In the case of women from dual-income households, the transition between work and family was positive when the husband helped with housework and parenting, and depression was alleviated [32,39]. In particular, marital intimacy was found to be a help among dual-income couples, with salutary associations with both wives’ work satisfaction and depression. Marital intimacy has been confirmed as an important factor positively affecting the mental health and work satisfaction of women from dual-income households [11,34,38,40,41]. Previous studies underpin this result. There is a representative system known as the parental leave system for work–family balance in Korean society, but it is considered that there is a need to provide more balanced work support systems and extra legal protection so that dual-income couples can manage their families together. In addition, specific psychological support measures should be provided within society and the workplace to maintain and enhance intimacy and resilience in dual-income couples. Above all, questioning the traditional division of roles along gender lines may be a step toward improving work and family balance in dual-income couples. Furthermore, actions that further gender equality and awareness within employment organizations should be promoted in the future and will have to be accompanied by active policy efforts by the states. 

This study has several limitations. First, only the sixth (2016) dataset from the Women and Families Panel Survey was utilized. Second, this study was limited to married, female wage earners: future studies should perform a comparative study by sub-classifying subjects according to their type of employment. Third, more comprehensive results would be obtained if the research were to address stress/depressive mood and perspectives on domestic labor in married men as well as in married women. Fourth, this study only measured the effects on depressive mood. It will be necessary to study other factors and mediated variables in future studies.

## 5. Conclusions

This study confirmed that husbands’ domestic labor is a factor that mediates mental health in women from dual-earner households via the cognitive and emotional satisfaction they experience at work and at home. This study has implications for the health sciences because women are generally the caregivers in the family and their health can affect the health of other family members. The basic data provided in this study may help enhance policies aimed at promoting and protecting women’s health. 

## Figures and Tables

**Figure 1 ijerph-17-04547-f001:**
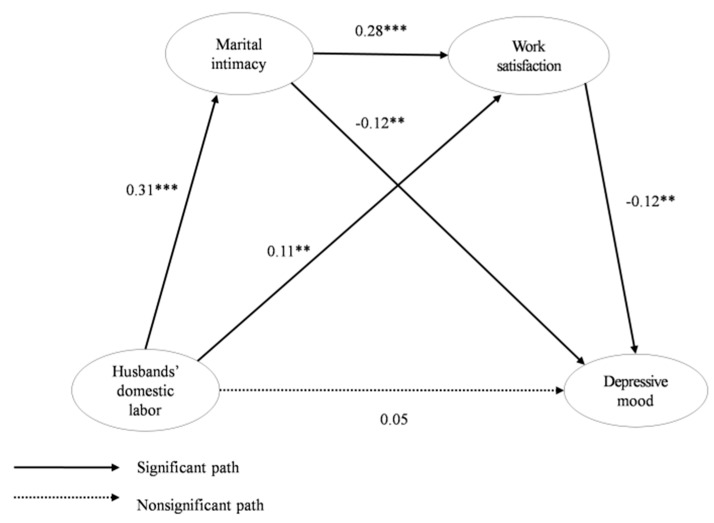
Final model. ** *p* < 0.01, *** *p* < 0.001.

**Table 1 ijerph-17-04547-t001:** General characteristics of participants (*n* = 791).

Characteristics	*N* (%)
**Age (Years)**	
≤39	142 (18.0)
40–49	425 (53.7)
50–59	183 (23.1)
60≤	41 (5.2)
**Education**	
≤High school	417 (52.7)
Associate degree	161 (20.4)
Bachelor’s degree	192 (24.3)
Master’s degree & Doctoral degree	21(2.7)
**Number of Children (≤18 Years)**	
none	266 (33.6)
1	191 (24.1)
2	281 (35.5)
3	52 (6.6)
4	1 (0.1)
**Wife’s Occupation**	
Professional work	243 (30.7)
Clerk	159 (20.1)
Service work	118 (14.9)
Sales work	93 (11.8)
Skilled agricultural, forestry, or fishery work	1 (0.1)
Craft or related trade work	22 (2.8)
Equipment, machinery operation, or assembly work	46 (5.8)
Elementary work	108 (13.7)
**Wife’s Employment Status**	
Regular	387 (48.9)
Irregular	404 (51.1)
**Wife’s Monthly Income (10,000 Won)**	
>100	111 (14.0)
100–199	426 (53.9)
200–299	141 (17.8)
300–399	57 (7.2)
400–499	35 (4.4)
500 ≤	21 (2.7)
**Husband’s Occupation**	
Manager	52 (6.6)
Professional work	93 (11.8)
Clerk	201 (25.4)
Service work	47 (5.9)
Sales work	33 (4.2)
Skilled agricultural, forestry, or fishery work	3 (0.4)
Craft or related trade work	222 (28.1)
Equipment, machinery operation, or assembly work	70 (8.8)
Elementary work	67 (8.5)
Soldier	3 (0.4)
**Husband’s Monthly Income (10,000 Won)**	
>100	7 (0.9)
100–199	92 (11.6)
200–299	210 (26.5)
300–399	248 (31.4)
400–499	103 (13.0)
500≤	131 (16.6)

**Table 2 ijerph-17-04547-t002:** Descriptive statistics.

Latent Variable	Measured Variable *	M	SD	Skewness	Kurtosis	Cronbach’s α
Husbands’ domestic labor	H1	2.38	1.46	0.914	−0.094	0.912
	H2	2.70	1.44	0.506	−0.606	
	H3	2.13	1.27	1.201	0.977	
	H4	2.39	1.11	0.785	0.871	
	H5	2.87	1.38	0.410	−0.448	
Work satisfaction	ws1	3.15	0.78	0.047	−0.238	0.928
	ws2	3.53	0.75	−0.055	−0.122	
	ws3	3.58	0.67	−0.208	0.032	
	ws4	3.54	0.68	−0.306	0.138	
	ws5	3.57	0.71	−0.389	0.266	
	ws6	3.28	0.78	−0.136	0.013	
	ws7	3.50	0.70	−0.111	0.148	
	ws8	3.13	0.85	−0.157	−0.404	
	ws9	3.25	0.74	−0.112	0.197	
	ws10	3.44	0.66	−0.046	0.083	
Depressive mood	dp1	0.35	0.60	1.706	2.812	0.897
	dp2	0.23	0.47	2.076	4.041	
	dp3	0.23	0.50	2.347	5.680	
	dp4	0.32	0.58	1.976	4.186	
	dp6	0.16	0.41	2.711	7.834	
	dp7	0.23	0.55	2.596	7.063	
	dp9	0.21	0.49	2.605	7.747	
	dp10	0.21	0.48	2.652	8.353	
Marital intimacy	mi1	2.90	0.63	−0.238	0.271	0.799
	mi2	2.80	0.61	−0.488	0.752	
	mi3	2.94	0.51	−0.637	2.650	
	mi4	3.11	0.50	−0.094	2.257	

* A detailed description of measured variables is provided in Appendix A. Abbreviations: M, mean; SD, standard deviation; H, husbands’ domestic labor; ws, work satisfaction; dp, depressive mood; mi, marital intimacy.

**Table 3 ijerph-17-04547-t003:** Correlation among latent variables.

	1	2	3	4
1. Husbands’ domestic labor	1			
2. Work satisfaction	0.188 ***	1		
3. Depressive mood	−0.036	−0.140 ***	1	
4. Marital intimacy	0.262 ***	0.278 ***	−0.134 ***	1

*** *p* < 0.001.

**Table 4 ijerph-17-04547-t004:** Confirmatory factor analysis.

Measured Variable		Latent Variable	Β	β	S.E.	C.R.	AVE	CR
H5	←	H	1.000	0.766			0.517	0.842
H4	←	H	0.906	0.864	0.036	24.933 ***		
H3	←	H	1.015	0.845	0.042	24.403 ***		
H2	←	H	1.093	0.800	0.042	25.873 ***		
H1	←	H	1.094	0.792	0.048	22.693 ***		
ws10	←	ws	1.000	0.871			0.706	0.960
ws9	←	ws	1.005	0.786	0.032	31.401 ***		
ws8	←	ws	1.017	0.685	0.046	22.117 ***		
ws7	←	ws	0.941	0.773	0.035	26.557 ***		
ws6	←	ws	1.063	0.782	0.039	27.058 ***		
ws5	←	ws	0.905	0.736	0.037	24.542 ***		
ws4	←	ws	0.930	0.789	0.034	27.317 ***		
ws3	←	ws	0.924	0.792	0.034	27.588 ***		
ws2	←	ws	0.866	0.666	0.041	21.278 ***		
ws1	←	ws	0.871	0.642	0.043	20.265 ***		
dp10	←	dp	1.000	0.737			0.800	0.969
dp9	←	dp	1.073	0.767	0.042	25.810 ***		
dp7	←	dp	0.927	0.599	0.058	15.945 ***		
dp6	←	dp	0.854	0.741	0.043	19.746 ***		
dp4	←	dp	1.147	0.693	0.063	18.200 ***		
dp3	←	dp	1.138	0.796	0.054	21.033 ***		
dp2	←	dp	1.030	0.766	0.051	20.342 ***		
dp1	←	dp	1.082	0.644	0.064	16.963 ***		
mi4	←	mi	1.000	0.652			0.767	0.929
mi3	←	mi	1.182	0.759	0.072	16.519 ***		
mi2	←	mi	1.212	0.650	0.082	14.846 ***		
mi1	←	mi	1.533	0.785	0.091	16.792 ***		

*** *p* < 0.001. Abbreviations: H, husbands’ domestic labor; ws, work satisfaction; dp, depressive mood; mi: marital intimacy; B, unstandardized coefficients; β, standardized coefficients; S.E., standard error; C.R., critical ratio; AVE: average variance extracted; CR, construct reliability.

**Table 5 ijerph-17-04547-t005:** Direct and indirect effects and mediated pathways.

Path				Β	β	S.E.	C.R.	p	95% C.I	
									**Lower**	**Upper**
Husbands’ domestic labor→			Marital intimacy	0.096	0.313	0.013	7.226	<0.001	0.071	0.125
Husbands’ domestic labor→			Work satisfaction	0.060	0.110	0.022	2.735	0.006	0.014	0.107
Husbands’ domestic labor→			Depressive mood	0.016	0.047	0.014	1.117	0.264	−0.018	0.054
Marital intimacy→			Work satisfaction	0.492	0.278	0.078	6.323	<0.001	0.326	0.696
Marital intimacy→			Depressive mood	−0.132	−0.122	0.051	−2.600	0.009	−0.258	−0.007
Work satisfaction→			Depressive mood	−0.076	−0.124	0.026	−2.956	0.003	−0.129	−0.029
Husbands’ domestic labor→	Marital intimacy→		Work satisfaction	0.492	-	0.078	6.323	<0.001	0.030	0.071
Husbands’ domestic labor→	Marital intimacy→		Depressive mood	−0.132	-	0.051	−2.600	0.009	−0.027	−0.001
Husbands’ domestic labor→	Work satisfaction→		Depressive mood	−0.076	-	0.026	−2.956	0.003	−0.011	−0.001
Husbands’ domestic labor→	Marital intimacy→	Work satisfaction→	Depressive mood	−0.076	-	0.026	−2.956	0.003	−0.007	−0.001

Abbreviations: C.I, confidence interval.

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
