# Peer review of "The Mediating Effects of Marital Intimacy and Work Satisfaction in the Relationship between Husbands’ Domestic Labor and Depressive Mood of Married Working Women"

_ijerph, 2020, doi:10.3390/ijerph17124547_

Round 1

Reviewer 1 Report

Husbands’ Domestic Labor Indirectly Effects the Depressive Mood of Married Working Women through Effects on Marital Intimacy and Work Satisfaction

The article builds a model analysing the effects of husbands’ housework participation on wives’ depressive moods. It has an innovative approach to analysing the mediating factors such as marital intimacy and work satisfaction.

However, the manuscript has many issues.

Major issues:

  • The positioning of the present paper within the wider housework and sociological literature is inadequate. There is no literature review, per se. For sociological research, this is not adequate. It is not clear what new findings and ideas this paper offers to the audience.
  • The discussion part, where some literature is brought about, is very disjoint. For instance, the first paragraph discusses the wage inequality and paid work participation of women, but the manuscript itself analyses husbands’ domestic labour, marital intimacy, job satisfaction, and depressive mood. No literature on the latter is discussed at all.
  • I cannot criticise the argument or the framework if neither is present in the paper. The manuscript needs a lot of work, particularly a thorough literature review in the area.

Minor issues:

  • The contribution of the paper to the current scientific knowledge should be made clear in the introduction
  • The title is confusing. Consider rephrasing it.
  • The dataset needs more elaboration. It is an interesting dataset, but not all are probably familiar with it. A little more detail (and proper citations) is necessary.

Author Response

Comments on Major issues

Answer) Thanks for your important comment on this study. As your comment of major issues, the authors confirmed once again whole contents, revised and added references.

This study aims to identify the relationship between husbands' household labor, marital intimacy, and work satisfaction with the depressive mood of dual-income women and to contribute to the policy and direction for promoting women's mental health.

The multi-role role of dual-income women can come as a potential risk factor for physical and mental health. The reality is not straightforward because housework and parenting are generally still considered the woman’s responsibility in Korea [1,2], and it is difficult to balance work and family [3,4].

Spouse is the most important source of social support and the first person to seek when faced with difficulties [7], and marital intimacy which is byproduct of marital interaction has been shown to reduce family and job stress by promoting the sharing of housework and parenting in dual earner couples [8], as well as stress about the role of work and family [9].

The women's wage inequality content was deleted from the first paragraph and revised to focus on the dual income situation of married women, as following.

"Dual earner households rate is high in their 40s to 50s when the cost of their children’s education is highly required, and it is explained by results of the Statistic Korea where women work mostly temporary or part-time [29,30]."

And the authors added the literature analysis on husband's housework, marital intimacy, work satisfaction, and depression to the latter as following.

“and this show the consistent result with previous studies showing that marital satisfaction increases as husbands’ domestic labor increases [33-35].”

“In a previous study of women in dual-earner, spouse support also revealed a static relationship with work satisfaction [36].”

“The level of depression of married working women is related to relationship with husband, satisfaction with their roles in family or workplace [39].”

“The marital intimacy is confirmed that an important factor in positively affecting mental health and work satisfaction of dual-income women [10,33,39,42,43].”

Minor issues:

  • The contribution of the paper to the current scientific knowledge should be made clear in the introduction

Answer) The authors described it on page 2 line 18.

  • The title is confusing. Consider rephrasing it.

Answer) As your comment, the authors revised the title.

“The Mediating Effects of Marital Intimacy and Work Satisfaction in the Relationship between Husbands’ Domestic Labor and Depressive Mood of Married Working Women.”

  • The dataset needs more elaboration. It is an interesting dataset, but not all are probably familiar with it. A little more detail (and proper citations) is necessary.

Answer) The data set used data from the sixth (2016) Women's Family Panel Survey conducted by the Korea Women's Development Institute (KWDI). This panel survey of 9,997 women aged 19 or older to 64 years younger was conducted every two years since 2007 (1st) and 2008 (2nd) and completed until the sixth round in 2016. The data was set married working women who do not have a housework assistant and who live with a wage-earner husband, including major variables was respond. And excluded non-responses and missing values.

Reviewer 2 Report

This manuscript describes a study of dual earner married women's depressive mood as predicted by husband's domestic labor, relationship satisfaction, and work satisfaction.  They authors used an existing dataset from 2016, which seems appropriate given the changes in women's participation in the workforce in Korea.  I had only a few comments:

1) The description of the dataset itself needs to be revised.  I couldn't follow how the data was collected and what inclusion and exclusion criteria were used to create the final sample. 

2) Was there missing data?  What did the authors do about the missing data.  I know that AMOS will not perform bootstrapping with missing data.

3)  The discussion of the CFA procedure seemed convoluted.  I would suggest simply saying you tested your measurement model and provide the fit of the original and the final model. 

4) The findings were similar to those found in the US and I assume other countries.  This was not referenced.  I think it would add to the manuscript if some of the literature from other countries was reviewed.

5) One of the limitations not mentioned was that you did not have the husbands' perspective on their domestic labor.  Research in the US would suggest that husbands and wives perspectives don't always match.

6) The authors mention trust and support in reference to husbands in the discussion section, but I was not clear what variables from their findings led them to that conclusion.           

Author Response

This manuscript describes a study of dual earner married women's depressive mood as predicted by husband's domestic labor, relationship satisfaction, and work satisfaction.  They authors used an existing dataset from 2016, which seems appropriate given the changes in women's participation in the workforce in Korea.  I had only a few comments:

Comment 1) The description of the dataset itself needs to be revised. I couldn't follow how the data was collected and what inclusion and exclusion criteria were used to create the final sample. 

Answer 1) The authors updated the detail of finding eligible population with exclusion criteria in the methods section. The updated results were 

“This panel survey of 9,997 women aged 19 or older to 64 years younger was conducted every two years since 2007 (1st) and 2008 (2nd) and completed until the sixth round in 2016. A total  of 9,997 KWDI 2016 subjects, those who rated as single, divorced, separation and bereavement (n=8,700), with housework assistant (n=89), not wage earner (n=412), and non-responses and missing values (n=5) were excluded. Finally, we selected survey responses from 791 married working women who live with a wage-earner husband and who do not have a housework assistant (Figure S1).”

In addition, the detail figure was shown in Figure S1 as selection eligible study population flow chart. The authors insisted reviewer 2 to check those updated contents with Figure S1.

2) Was there missing data?  What did the authors do about the missing data.  I know that AMOS will not perform bootstrapping with missing data.

Answer 2) As you commented, AMOS was unable to perform bootstrapping with missing data, so we excluded the missing data from the study design. According to the final exclusion for missing value level, the authors found only 5 subjects with missing values. Therefore, the authors decided to delete those missing values to make the result more clearly, not to use reputation the missing values.

3)  The discussion of the CFA procedure seemed convoluted.  I would suggest simply saying you tested your measurement model and provide the fit of the original and the final model. 

Answer 3) Thanks for your important suggestion on CFA procedure. To help understanding the validity and fitness verification of the measurement model, it is written in several journals, and so is this study, as described following.

“First, to verify the validity of the measurement model, confirmatory factor analysis of potential variables was conducted to confirm the convergent validity, discriminant validity, and the fitness index [23,24].
Next, the fitness of the structural equation model was verified using the maximum likelihood method. χ², SRMR, RMSEA, and GFI were used for the absolute fit index; NFI, TLI, and CFI were used for the incremental fit index; and PRATIO, PNFI, and PCFI were used for the parsimonious fit index [24].”

4) The findings were similar to those found in the US and I assume other countries.  This was not referenced.  I think it would add to the manuscript if some of the literature from other countries was reviewed.

Answer 4) As your comment, the authors added the 2 paper for reference 39, 41 on discussion section.

5) One of the limitations not mentioned was that you did not have the husbands' perspective on their domestic labor.  Research in the US would suggest that husbands and wives perspectives don't always match.

Answer 5) Thanks for your important comments on limitation section. As your comment, Page 10 Line 2 changed as Thanks for your important comments on limitation section.
As your comment, Page 10 Line 3 changed as “Third, more comprehensive results would be obtained if the research addressed stress, depression and perspective on domestic labor in married men as well as in married women.”

6) The authors mention trust and support in reference to husbands in the discussion section, but I was not clear what variables from their findings led them to that conclusion.         

Answer 6) The sentence was removed on discussion section, and then was described as following;

“In the case of dual-income women, the transition between work and family was positive when the husband helped with housework and parenting, and depression was alleviated [39,40]. In particular, marital intimacy was found to be a help among dual-income couples, with salutary associations with both wives’ work satisfaction and depression. The marital intimacy is confirmed that an important factor in positively affecting mental health and work satisfaction of dual-income women [10,33,38,41].”

The authors highlighted those changes in the manuscript. We also appreciate you sincere comment on the present study.

Reviewer 3 Report

Overall I enjoyed reading this paper and believe it has useful data, worth publishing.  I am a psychologist, so I looked at the manuscript from that perspective.  I realize what is expected in other disciplines may differ.

The authors write:

This study aims to analyze the causal relationships between the amount of housework performed by the husband, marital intimacy, and work satisfaction in married working women. These are anticipated to have important effects on the depressive mood of married working women who are required to play multiple roles at work and at home. The overall purpose of this study is to provide the basic data necessary to prepare a women's health policy aimed at enhancing the psychological well-being and mental health of married working women.

As I read the authors statement of their goals and look at their Figure 1, I infer they are implying that husbands’ domestic labor contributes to marital intimacy and work satisfaction which in turn buffer wives from depression.  This seems plausible to me but it also seems plausible to me that depression undermines marital intimacy and work satisfaction.  At some point in their paper, probably the discussion, I recommend that the authors note the issue of inferring causal direction from cross-sectional data.

I feel the introduction to the study would benefit from some discussion of past research on the same or similar variables and a justifications of what I see as the authors’ predictions. I am pasting below a couple of leads on potentially relevant sources. 

With the authors’ goal of informing policy, there are two aspects of the division of household labor that are noticeable to me.  a) Overall the predictive strength of the relationships among the variables in this study is low (i.e., correlations less than .30).  b) The division of household labor is not something such as seatbelt laws that governments can easily change.  So I would welcome the authors noting and commenting upon the relatively weak predictive power of the variables, the value of using weak effects as the basis of policy recommendations, and the kinds of policy changes they believe might follow from their results. 

Page 2, line 65ff: “They are wage workers, including regular, temporary and daily employees [4], characterized time use is inelastic than self-employed workers and non-wage earners.”  Were words omitted in this passage? 

Page 3, line 123ff: “Demographic characteristics are shown in Table 1. The major variables had skewness values ranging from -0.637 to 2.711 and kurtosis values ranging from -0.606 to 8.353, and thus satisfied the assumptions for normality.”  Some of these scores do not seem to me to have normal distributions.  For example, I would expect the distribution of CES-D scores to be skewed.

As a reader I appreciate being able to see a matrix of correlations among variables.  I would encourage the authors to include Table S1 in the manuscript per se. 

In the bootstrap analysis, I would indicate the number of repetitions run.

Page 9, penultimate paragraph: “The support and trust of the spouse was found to be a great help among dual-income couples, with positive effects on both physical and psychological health as well as the marital relationship.” The marital intimacy measure included a trust item but it was not used as a single item indicator in the analysis.  Given the analyses performed in this study, I believe a more precise statement might be something such as “Marital intimacy was found to be a help among dual-income couples, with salutary associations with both wives’ work satisfaction and depression.”

The way I use the notion of a mediating variable, mediating variables are intervening processes. Thus, the mediating variables in the way I understand this study are marital intimacy and work satisfaction, not husbands’ domestic labor.  This affects the phrasing I would use at various points in the manuscript (e.g., the abstract, the last paragraph).  I see the husbands’ domestic labor as what I might call the “independent” or the “predictor” variable.

In sum, I am pleased to see the work the authors have done.  At least for psychologically trained readers, I feel the authors should provide a more thorough use of existing body of related work as a way contextualizing this study and justifying their expectations about the result.  I also hope the authors will talk more about the magnitude of the correlations, the direction of causality, and the policy implications they see based on their findings. 

Fincham, F. D., Beach, S. R., Harold, G. T., & Osborne, L. N. (1997). Marital satisfaction and depression: Different causal relationships for men and women?. Psychological Science8(5), 351-356.

Bird, C. E. (1999). Gender, household labor, and psychological distress: The impact of the amount and division of housework. Journal of Health and Social behavior, 40, 32-45. [Depression was used as the distress measure]

Oshio, T., Nozaki, K., & Kobayashi, M. (2013). Division of household labor and marital satisfaction in China, Japan, and Korea. Journal of Family and Economic Issues34(2), 211-223.

Qian, Y., & Sayer, L. C. (2016). Division of labor, gender ideology, and marital satisfaction in East Asia. Journal of Marriage and Family78(2), 383-400.

Saginak, K. A., & Saginak, M. A. (2005). Balancing work and family: Equity, gender, and marital satisfaction. The Family Journal13(2), 162-166.

Michalos, A. C. (2003). Job satisfaction, marital satisfaction and the quality of life: A review and a preview. In Essays on the quality of life (pp. 123-144). Springer, Dordrecht.

Author Response

This study aims to analyze the causal relationships between the amount of housework performed by the husband, marital intimacy, and work satisfaction in married working women. These are anticipated to have important effects on the depressive mood of married working women who are required to play multiple roles at work and at home. The overall purpose of this study is to provide the basic data necessary to prepare a women's health policy aimed at enhancing the psychological well-being and mental health of married working women.

Comment 1) I feel the introduction to the study would benefit from some discussion of past research on the same or similar variables and a justifications of what I see as the authors’ predictions. I am pasting below a couple of leads on potentially relevant sources. 

 Answer 1) Thank you for your comments and useful literature. The authors referred to the literature and added the 3 papers for reference 3,4,6.

Comment 2) With the authors’ goal of informing policy, there are two aspects of the division of household labor that are noticeable to me.  a) Overall the predictive strength of the relationships among the variables in this study is low (i.e., correlations less than .30).  b) The division of household labor is not something such as seatbelt laws that governments can easily change. So I would welcome the authors noting and commenting upon the relatively weak predictive power of the variables, the value of using weak effects as the basis of policy recommendations, and the kinds of policy changes they believe might follow from their results. 

Answer 2) Thanks for your comments. Division of household labor of spouse is valuable as a highly important factor leading a balance and satisfaction of healthy family and work life to dual-income women. The author described the policy on page 9, line 39 and below in the discussion section.

Comment 3) Page 2, line 65ff: “They are wage workers, including regular, temporary and daily employees [4], characterized time use is inelastic than self-employed workers and non-wage earners.”  Were words omitted in this passage? 

 Answer 3) Through a detailed review, the authors were confirmed once again. Page 2 Line 16 changed as "This study designed for wage-earner, who account for 77.2% of all female wage workers, including regular, temporary and daily employees [4]."

Comment 4) Page 3, line 123ff: “Demographic characteristics are shown in Table 1. The major variables had skewness values ranging from -0.637 to 2.711 and kurtosis values ranging from -0.606 to 8.353, and thus satisfied the assumptions for normality.”  Some of these scores do not seem to me to have normal distributions.  For example, I would expect the distribution of CES-D scores to be skewed.

Answer 4) As your comment, the skewness and kurtosis values were included within the standard, but some of the data could not satisfy the normality assumption.

Normality is an issue because it is one of the basic assumptions required in order to carry out structural equation modelling (SEM) analysis (Byrne, B. M. 2010), and skewed estimates may appear if each measurement variable is not normally distributed, so the skewness and kurtosis values of the major variables was analyzed beforehand to verify the normality of the data (Keum, J.; Kim, D. 2014).

Byrne, B. M., Structural equation modeling with AMOS: Basic concepts, applications, and programming, 2nd ed. Routledge/Taylor & Francis Group: New York, NY, US, 2010; pp xix, 396-xix, 396.

Keum, J.; Kim, D. The Casual Relationship among the Father’s Participation in Childcare, Job Satisfaction, Parenting Stress, and Marital Satisfaction of Working Mother. Fam. Environ. Res 2014, 52, 141-150, doi:10.6115/fer.2014.52.2.141.

Comment 5) As a reader I appreciate being able to see a matrix of correlations among variables.  I would encourage the authors to include Table S1 in the manuscript per se. 

 Answer5) As your comment, the authors added the correlation table (Table 3) of this paper.

Comment 6) In the bootstrap analysis, I would indicate the number of repetitions run.

Answer 6) The authors were described in the statistical analysis section as following; "Since the accuracy of bootstrapping increases with the number of estimates, the number of estimates was set to 10,000."

Comment 7) Page 9, penultimate paragraph: “The support and trust of the spouse was found to be a great help among dual-income couples, with positive effects on both physical and psychological health as well as the marital relationship.” The marital intimacy measure included a trust item but it was not used as a single item indicator in the analysis.  Given the analyses performed in this study, I believe a more precise statement might be something such as “Marital intimacy was found to be a help among dual-income couples, with salutary associations with both wives’ work satisfaction and depression.”

 Answer 7) Thanks for your important comments on discussion section. The authors changed it as following; "Marital intimacy was found to be a help among dual-income couples, with salutary associations with both wives’ work satisfaction and depression.”

Comment 8) The way I use the notion of a mediating variable, mediating variables are intervening processes. Thus, the mediating variables in the way I understand this study are marital intimacy and work satisfaction, not husbands’ domestic labor.  This affects the phrasing I would use at various points in the manuscript (e.g., the abstract, the last paragraph).  I see the husbands’ domestic labor as what I might call the “independent” or the “predictor” variable.

Answer 8) This study designed husbands' domestic labor as an independent variable, marital intimacy and work satisfaction as mediating variables, and dual-income women's depressive mood as dependent variable.

The authors highlighted those changes in the manuscript. We also appreciate you sincere comment on the present study.

Round 2

Reviewer 1 Report

This is a substantial improvement from the previous version. The study makes a clear contribution to the explanation of the processes behind the association between husbands’ housework participation and wives’ depressive moods. Beyond the listed contributions to health policies and the psychological research of depression, it would be really wonderful if you could bridge in the sociological research on the issue. Particularly, the potential academic (and theoretical) impact of the paper would increase with a more in-depth dialogue with Qian and Sayer (2016).

Author Response

Dear Reviewer 1

I and all of authors in the present totally agree your opinion on sociological research with depressive mood.

There are several limitation on elucidation of the conclusion and its implementation on psychological health issue with controlling potential confoundings and bias.

The authors tried to show the possible implementation at the end of the manuscript. We tried to write those section within our results not to induce flight of idea. 

We really appreciate your sincere comment for updating quality of the present study.

Best regards

Reviewer 3 Report

  1. Overall the authors have conscientiously considered and endeavored to address the points raised in the first round of reviews. I appreciate their efforts. I continue to feel this paper has reasonable data and can be published.
  1. The authors appear to agree with me that some of their data is non-normally distributed: in their response they write “some of the data could not satisfy the normality assumption.” Nonetheless in the paper itself they have left the phrasing “thus satisfied the assumptions for normality.” I might phrase this: “thus did not, for every variable, satisfy the assumptions for normality.”
  1. The authors’ first change was to rephrase the title. I hadn’t specifically commented on that but their revised title is related to a point I made in my review regarding which were the mediating variables. I like that the new title complements the point I was making.
  1. On page 2 the authors made changes to the phrasing of the second sentence in the paragraph on the study population and added new text. I feel the phrasing of mid-section of that paragraph could be further polished.

This Panel study, initially started in 2007, was designed to examine female wage-earners, including regular, temporary and daily employees, who account for 77.2% of all female workers [12]. The data used in the present report were from the sixth (2016) wave.  That wave consisted of 9,997 women, 19 to 64 years old. Of the total sample, participants were excluded from the current analyses if: they were single, divorced, separated, or widowed (n=8,700); had a housework assistant (n=89); were married to a husband who was not a wage earner (n=412); and/or who had non-responses and missing values (n=5).

  1. Thanks for including the correlation table.
  1. My goof for not seeing that in their first draft the authors indicated the number of bootstrap estimates was set to 10,000.
  1. On page 9, I was pleased the authors followed up on the possible rephrasing of the sentence that now begins “In particular, marital intimacy was found ….”.

Author Response

Dear Reviewer 3

I as correspond author on behalf of whole authors appreciate you of your sincere comments on the present study.

The authors reviewed you comments, the reply on your comments was summrised and described as following;

Comment) The authors appear to agree with me that some of their data is non-normally distributed: in their response they write “some of the data could not satisfy the normality assumption.” Nonetheless in the paper itself they have left the phrasing “thus satisfied the assumptions for normality.” I might phrase this: “thus did not, for every variable, satisfy the assumptions for normality.”

reply) As your comment, changed the sentence on page 4 line 4 as following;

“, but thus did not, for every variable, satisfy the assumptions for normality.”

Comment) On page 2 the authors made changes to the phrasing of the second sentence in the paragraph on the study population and added new text. I feel the phrasing of mid-section of that paragraph could be further polished.

reply)

There was a misunderstanding about the panel survey, so the authors was changed and corrected the position of sentence.

“The data of study used in the present report were from the sixth(2016) wave of the the panel study, initially started in 2007 by the Korea Women's Development Institute (KWDI). That wave consisted of 9,997 women, 19 to 64 years old. The subjects of study were designed for female wage-earners, including regular, temporary and daily employees, who account for 77.2% of all female workers in Korea [12]. Of the total sample, participants were excluded from the current analyses if: they were single, divorced, separated, or widowed(n=8,700); had a housework assistant (n=89); were married to a husband who was not a wage earner (n=412); and/or who had non-responses and missing values(n=5) (Figure S1).”

Please follow the point by point reply above.

Best regards